# Deep Denoising Prior: You Only Need a Deep Gaussian Denoiser

## Abstract

Gaussian denoising often serves as the initiation of research in the field of image denoising, owing to its prevalence and intriguing properties. However, deep Gaussian denoiser typically generalizes poorly to other types of noises, such as Poisson noise and real-world noise. In this paper, we reveal that deep Gaussian denoisers have an underlying ability to handle other noises with only ten iterations of self-supervised learning, which is referred to as *deep denoiser prior*. Specifically, we first pre-train a Gaussian denoising model in a self-supervised manner. Then, for each test image, we construct a pixel bank based on the self-similarity and randomly sample pseudo-instance examples from it to perform test-time adaptation. Finally, we fine-tune the pre-trained Gaussian denoiser using the randomly sampled pseudo-instances. Extensive experiments demonstrate that our test-time adaptation method helps the pre-trained Gaussian denoiser rapidly improve performance in removing both in-distribution and out-of-distribution noise, achieving superior performance compared to existing single-image denoising methods while also significantly reducing computational time.

## 1 Introduction

Image denoising is a crucial task in the field of computer vision, aimed at restoring the clean image from noisy input. Most previous denoising methods (Liang et al., 2021; Lehtinen et al., 2018; Ryou et al., 2024; Ai et al., 2024) assume that noisy images follow a specific noise distribution. However, in real-world noisy images (Xu et al., 2018; Abdelhamed et al., 2018; Zhang et al., 2019), the type

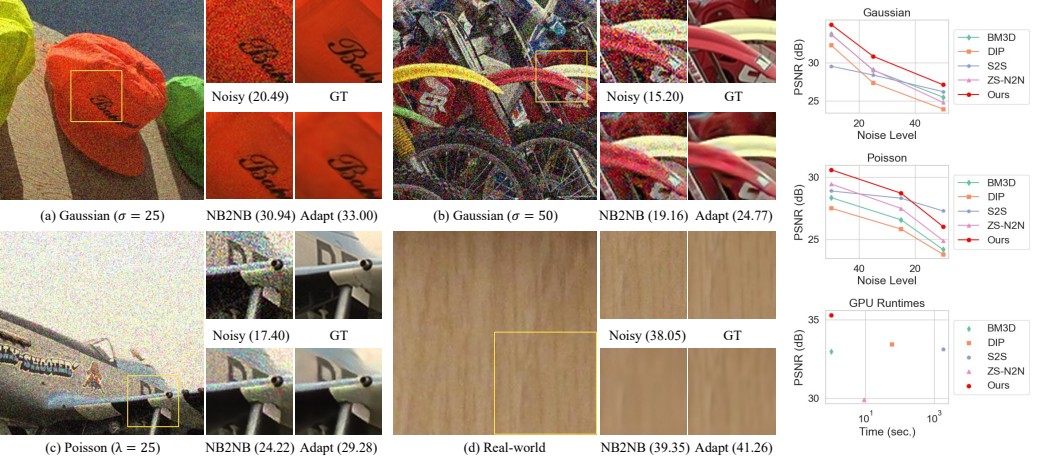

Figure 1: Through the proposed test-time adaptation strategy, we enable the pre-trained Gaussian denoiser to rapidly adjust to the unknown noise characteristics of the test images, allowing for more accurate denoising across varying noise distributions. NB2NB is trained on Gaussian noise with $\sigma = 25$ and tested on Gaussian noise with $\sigma = 25$ and 50, Poisson noise with $\lambda = 25$, and real-world noise. On the right, from top to bottom, are the performances of different dataset-free image denoising methods on Gaussian noise, Poisson noise, and real-world noise against test time.

and level of noise are unknown and often differ from those in the training images, thereby limiting the effectiveness of pre-trained denoising models. To alleviate this issue, several denoising methods (Jin et al., 2020; Park et al., 2009; Lee et al., 2022; Pan et al., 2023; Fu et al., 2023; Vaksman & Elad, 2023) targeting real-world noisy images have been proposed, improving the performance of denoisers on real-world noise. However, they also require training on large datasets of real-world noisy images with the same noise distribution, which greatly limits their practicality.

Recently, single-image denoising methods (Ulyanov et al., 2018; Quan et al., 2020; Mansour & Heckel, 2023; Chihaoui & Favaro, 2024) have improved practicality by training an image-specific denoising model for each test image, eliminating the dependence on specific training data. The Self2Self network (Quan et al., 2020) performs dropout training on Bernoulli-sampled instances of the input image, effectively preventing overfitting and achieving good denoising performance. Noise2Fast (Lequyer et al., 2022) introduces checkerboard downsampling, generating four low-resolution images and training a smaller network with eight convolutional layers to increase speed at the expense of accuracy. ZS-N2N (Mansour & Heckel, 2023) trains a two-layer network on two downsampled images to further improve the speed of algorithm. Clearly, the trend in existing single-image denoising methods is to enhance speed by reducing the network size; however, this reduction inevitably limits the performance of the algorithm, and ZS-N2N has already minimized the network to two layers, making further reductions impractical. Moreover, these methods only leverage prior information from a single noisy image, which restricts the performance of the denoiser.

Test-time adaptation (TTA) has recently shown considerable success in various fields, including sentiment analysis and machine translation in natural language processing (Shu et al., 2022; He et al., 2021), as well as object detection and image classification in computer vision (Hsu et al., 2020; Wang et al., 2021). TTA allows the model to be fine-tuned during the inference phase, enabling rapid adaptation to the distribution of new input data, thereby improving accuracy and robustness. SS-TTA (Fahim & Boutellier, 2023), for the first time, applies test-time adaptation to the image denoising task, enhancing the performance of self-supervised denoising methods. However, SS-TTA requires the noise type of the test images to match the noise type used in the training images for the pre-trained denoising model, such as both being Gaussian noise. When the noise type of the test image is unknown, SS-TTA may fail to learn how to remove the unknown noise from the test image. Therefore, designing a test-time adaptation denoising method that is independent of the noise distribution in the test images remains an open problem.

In this study, we address the challenge of balancing speed and performance in single-image denoising from a novel perspective. We focus on Gaussian denoising models in the era of deep learning. Due to the significant distribution difference between training data and test data, denoising models trained on a specific Gaussian noise distribution perform poorly when faced with other types of noise, lacking generalization within the same distribution and transferability across distributions. It is important to note that purely Gaussian denoising models may still hold some efficacy in mitigating out-of-distribution noise, even if they are not specifically designed for such scenarios. In traditional image denoising, the introduction of variable splitting techniques allowed Gaussian models to act as plug-and-play denoisers in iterative algorithms designed to handle various types of noise (Venkatakrishnan et al., 2013). This concept has also been considered in the deep learning era (Ryu et al., 2019). In this paper, we refrain from considering models trained on large-scale image data solely as denoisers; rather, we interpret them as models that encapsulate rich image priors, given their extensive exposure to diverse images. We find that while deep learning models are significantly affected when the noise in the input differs from the training data, with appropriate fine-tuning, the denoisers can quickly adapt to test images with different noise distributions.

To this end, we propose a test-time adaptive denoising model. First, we pre-train a denoiser in a self-supervised manner under a predefined Gaussian noise distribution. Then, based on self-similarity, we adapt the network by constructing a *pixel bank* from samples of the test image, allowing rapid adaptation to the target domain. Specifically, we leverage the non-local self-similarity of the image, searching for similar pixels within a sufficiently large window to build the pixel bank. At the same time, we generate pseudo-instances using a tailored pixel-wise random sampling strategy to adapt the denoising network. This strategy facilitates swift convergence, allowing the network to achieve high-quality denoising in just a few iterations (*e.g.*, 10 iterations), whereas existing zero-shot denoising methods often require thousands of iterations. Experimental results demonstrate that the proposed method improves the denoising performance of the pre-trained model for both in-distribution and out-of-distribution noise. Compared to existing single-image denoising methods,

our approach exhibits significant advantages in both denoising performance and runtime efficiency (as shown in Figure 1). Our main contributions are summarized as follows:

- We propose a test-time adaptive denoising framework, which adapts the pre-trained denoising model to different target domains during testing. Without accessing any real clean images, the proposed TTA denoising framework can be applied in a self-supervised manner to real-world scenarios with unknown noise distributions.

- We construct a pixel bank based on the self-similarity prior of the test image and sample pseudo-instances from it to perform adaptive fine-tuning of the network. Our approach can quickly adapt to different noise distributions.

- We demonstrate that Gaussian denoisers pre-trained on natural images possess a deep denoising prior, which can be quickly adapted through fine-tuning to remove other types of noise. The proposed method outperforms existing single-image denoising approaches in terms of both denoising performance and runtime efficiency.

## 2 PRELIMINARIES

The modeling of image corruption can be described using a simple probabilistic framework: Given a clean image $x$, the observed noisy image $y$ can be regarded as drawn from the noise model $p(y|x)$.

For example, additive Gaussian white noise can be represented as $p(y|x) = \frac{1}{\sqrt{2\pi}\sigma}e^{-\frac{\|y-x\|^2}{2\sigma^2}}$. Existing denoiser training approaches mainly fall into two categories: training networks with noisy-clean image pairs and training networks with noisy-noisy image pairs.

**Noise2Clean training.** Noise2Clean paradigm has access to both noisy and clean images, which aims at end-to-end training of a denoising model $f(\cdot)$[1] with noisy-clean image pairs. With a large number of pairs $(x, y)$, we usually minimize the empirical risk to train the model $f$, and the corresponding expected risk $R_L(f)$ is defined as

$$R_L(f) = \mathbb{E}_{x,y}L(f(y), x), \tag{1}$$

where $L$ denotes the loss function.

**Noise2Noise training.** In many real-world scenarios, the ground-truth clean images are physically unavailable. Noise2Noise (Lehtinen et al., 2018) proposes to leverage noisy-noisy image pairs from the same scene in place of noisy-clean pairs, which can remarkably attain performance nearly equivalent to the latter. With pairs of noisy images $(y, z)$, the Noise2Noise training learns a denoiser by minimizing the expected risk defined as follows (Zhou et al., 2023):

$$R'_L(f) = \mathbb{E}_{y,z}L(f(y), z) = \mathbb{E}_y\mathbb{E}_{x|y}\mathbb{E}_{z|x}L(f(y), z) = \mathbb{E}_{x,y}\left[\mathbb{E}_{z|x}L(f(y), z)\right], \tag{2}$$

where the second equality is based on the fact that $p(z|y) = \int p(z|x)p(x|y)\mathrm{d}x$, and $p(z|x)$ denotes the probability of corrupting the clean image $x$ into the noisy image $z$. Eq. 2 shows that as long as $z$ and the loss function $L$ possess certain properties (e.g., $z$ has zero-mean noise and $L$ is an $\ell_2$ loss), the minimization of noise2noise training risk is equivalent to minimizing noise2clean training risk.

**Domain shift between training and testing.** Existing dataset-based image denoising methods (Zhang et al., 2017; Zhao et al., 2023; Zhou et al., 2024) typically construct paired noisy-clean training images by synthesizing noisy images from clean ones using predefined noise distributions (e.g., Gaussian distribution) or using real-world noisy images collected from actual environments to train denoisers. However, due to the diversity of camera sensors and unknown processing on the internet, the noise distribution of real-world test images may differ from that of the training images—a phenomenon known as domain shift (Koh et al., 2021; Wang et al., 2022a). In these cases, denoising models often fail to produce satisfactory clean images.

## 3 METHOD

In this section, we demonstrate that noise-agnostic image denoising can be greatly improved with *test-time adaptation* in the following three steps:

---

[1]In deep learning, $f$ is usually a neural network parameterized by $\theta$.

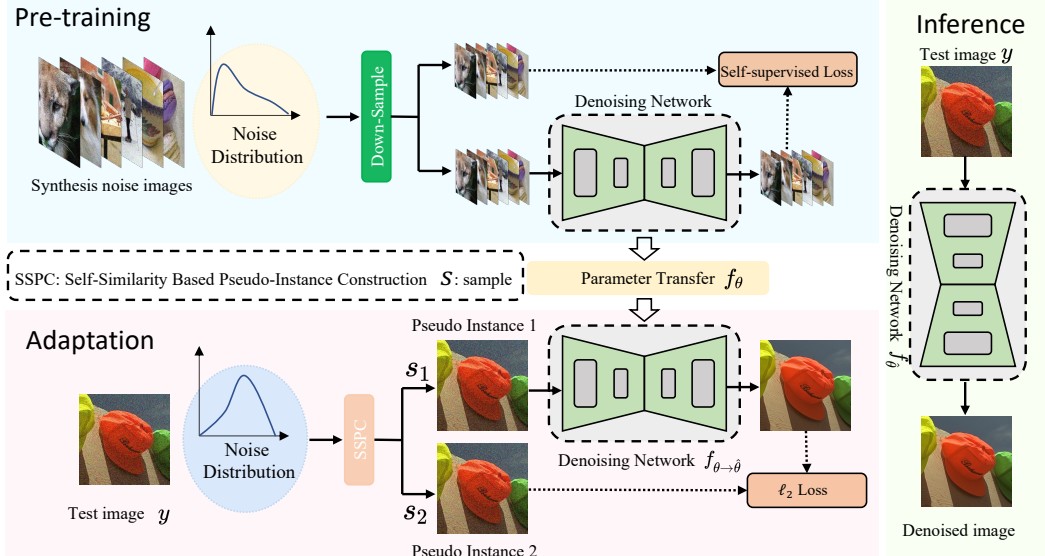

Figure 2: **Overview of the proposed Test-Time Adaptive Denoising framework.** Pre-training: Self-supervised training using synthetic Gaussian noisy images. Adaptation: Constructing pseudo-instances from test noisy images to fine-tune the network. Before each iteration, we sample two pseudo instances through the SSPC. Inference: Inference using the fine-tuned denoising network and the original noisy image.

**Step 1.** *Preparing a pre-trained Gaussian denoising model*

From traditional approaches to modern deep learning methods, Gaussian denoising has long been a starting point in image denoising research due to its particularly favorable mathematical properties. In this work, we focus on Gaussian denoising models in the deep learning era, which can typically leverage large amounts of training data through cost-effective methods. However, a model trained exclusively on Gaussian noise performs poorly on other types of noise, due to significant distributional differences between training and testing data, lacking both in-distribution generalization and cross-distribution transferability. Therefore, it is necessary to construct noisy-noisy image pairs from the test images to fine-tune the model. While deep learning models are often highly susceptible to noise contamination from unfamiliar sources, they can be steered away from this interference through simple methods, leading to significant improvements in denoising performance for out-of-distribution noise.

**Step 2.** *Building noisy-noisy image pairs from the noisy image to be denoised*

To achieve test-time adaptation when clean reference images are unavailable, it is essential to construct training data using the noisy test image. High-quality training data is critical for fine-tuning the network, especially in single-image denoising tasks. Carefully constructed noise-noise image pairs can accurately capture the noise characteristics of the target image, enabling the pre-trained Gaussian denoising model to adapt more effectively to new noise distributions during fine-tuning. This not only improves the model's performance in removing specific noise types but also enhances its generalization across various noise conditions. Most existing methods generate training data pairs through downsampling, which presents several drawbacks, such as overfitting due to the limited amount of training data and a significant similarity gap in the clean content of the noisy training images. We believe that self-supervised image denoising relies on a crucial implicit assumption (or prior): the pervasive presence of self-similarity in images. Therefore, we construct training data based on the inherent self-similarity of images.

**Step 3.** *Fine-tuning the pre-trained Gaussian denoising model with the Noise2Noise framework*

After preparing the noise-noise image pairs, we fine-tune the pre-trained Gaussian denoising model using the Noise2Noise framework. This approach leverages the insight that the model can learn to recover the underlying clean signal by mapping between different noise instances within the same image. As a result, the model quickly adapts to the noise distribution of the test image, enabling the recovery of high-quality images.

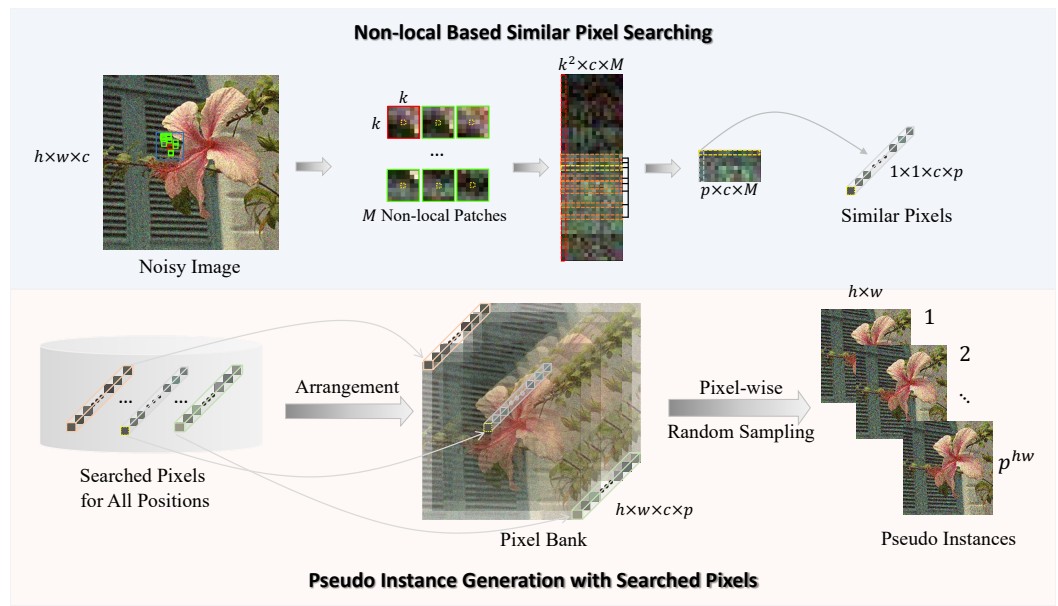

Figure 3: **Overview of the proposed Self-Similarity Based Pseudo-Instance Construction framework.** Top: Non-Local Similar Pixel Search. For each pixel in the noisy image, we crop a local patch of size $k \times k$ centered at that pixel, then search for $m$ non-local patches and sort them according to similarity. Rearrange the pixels of each non-local patch into a column and assemble the $m$ non-local patches into a matrix. Search for the $p$ rows that are most similar to the middle row of the matrix, sort them, and extract the first pixel from each row. Bottom: Generating Pseudo-Instances Using Searched Pixels. Arrange all the pixels found on the left into a pixel library tensor. During network training, randomly sample two pseudo-instances per pixel from this tensor in each iteration to form the input and output of the network.

Next, we will provide a detailed explanation of the specific methods and implementation details of these three steps.

### 3.1 PRE-TRAIN GAUSSIAN DENOISING MODEL

We pre-train a Gaussian denoiser in a self-supervised manner, as illustrated in Figure 2. Since network design is not the focus of this work, we adopt the same architecture and training methods as those used in Neighbor2Neighbor (Huang et al., 2021). We then treat the pre-trained Gaussian denoiser as a model with sufficient image priors, aiming to quickly adapt it to both in-distribution and out-of-distribution (OOD) noise using a straightforward approach.

We attempt to use the data construction method from the existing single-image denoising approach, ZS-N2N, to generate noisy image pairs and fine-tune the pre-trained Gaussian denoiser. The results, shown in Table 1, are based on a Gaussian denoiser pre-trained on Gaussian noise with a noise level of $\sigma = 25$. In the following experiments, we use the same pre-trained model. Fine-tuning and testing are conducted on Gaussian noise with $\sigma = 25, 50$ and Poisson noise with $\lambda = 10, 25$. When fine-tuning with the pre-trained model, we perform 10 iterations, whereas, without it, the network requires 2000 iterations. As shown in the table, using the pre-trained Gaussian denoiser as the initialization model not only significantly accelerates the convergence speed of ZS-N2N but also improves its performance for both in-distribution and out-of-distribution (OOD) noise. This demonstrates that the Gaussian denoising model, trained on large datasets, possesses priors—what we refer to as the "deep denoising prior".

To further investigate the impact of different data construction methods on the performance of pre-trained denoising models, we used the data construction approaches from existing self-supervised denoising methods, NB2NB and ZS-N2N, to generate noisy image pairs for fine-tuning the pre-trained Gaussian denoiser. The results, presented in Table 2, in most cases, the network's denoising performance improved after fine-tuning. However, for ZS-N2N at $\sigma = 25$ (in-distribution noise), fine-tuning resulted in a decline in performance. Furthermore, the two fine-tuning methods exhib-

Table 1: The average PSNR scores and runtime of ZS-N2N with and without test-time adaptation for Gaussian and Poisson denoising on Kodak24 dataset.

| Method | Gaussian | | Poisson | | time (s) |
|---|---|---|---|---|---|
| | $\sigma$=25 | $\sigma$=50 | $\lambda$=25 | $\lambda$=10 | |
| ZS-N2N (w/o) | 29.07 | 24.81 | 27.49 | 24.92 | 9.04 |
| ZS-N2N (w) | **29.33** | **26.07** | **28.65** | **25.86** | 0.07 |

Table 2: The pre-trained Gaussian denoiser employs different downsampling data methods for test-time adaptation.

| Method | Gaussian | | Poisson | |
|---|---|---|---|---|
| | $\sigma$=25 | $\sigma$=50 | $\lambda$=25 | $\lambda$=10 |
| Pre-trained | 29.70 | 20.94 | 24.28 | 19.94 |
| NB2NB | **30.35** | 25.07 | **28.69** | **25.86** |
| ZS-N2N | 29.33 | **26.07** | 28.65 | **25.86** |

ited different strengths across various noise distributions. This suggests that the method used for constructing fine-tuning data significantly influences the network's denoising performance, and selecting an inappropriate method can lead to performance degradation. Therefore, to fully leverage the deep denoising prior of the pre-trained Gaussian denoiser, an effective training data generation method based on the test image must be designed.

### 3.2 BUILDING NOISE2NOISE PAIRS VIA PIXEL SEARCH

Given a noisy image $y \in \mathbb{R}^{h \times w \times c}$, we need to construct Noise2Noise training data pairs from it. A key point here is the similarity of the clean content in the training data pairs. When the clean content similarity is high, minimizing the Noise2Noise training loss becomes approximately equivalent to minimizing the Noise2Clean training loss; otherwise, significant errors may be introduced. Previous methods primarily employed various downsampling techniques to create training data, resulting in size inconsistencies between the training and test data. Moreover, each method has its own limitations. For instance, Neighbor2Neighbor (Huang et al., 2021) and Blind2Unblind (Wang et al., 2022b) sample similar pixels within a $2 \times 2$ region, but in regions of the image where the content changes significantly (such as corners and lines), there may not be similar pixels in such a small region. Noise2Fast (Lequyer et al., 2022) and ZS-N2N (Mansour & Heckel, 2023) generate only four and two low-resolution images, respectively. To avoid overfitting, they can only train very small networks, which limits the overall performance.

To generate training data with higher clean content similarity for fine-tuning the pre-trained Gaussian denoising model, we leverage the self-similarity widely present in natural images to search for similar pixels and construct data to fine-tune. Next, we introduce a simple and effective method that fully utilizes self-similarity prior of the test image. The overall process of this method is illustrated in Figure 3. For simplicity in description and illustration, we omit the channel dimension $c$ when describing images. For example, we represent $k^2 \times c \times M$ as a matrix and $1 \times 1 \times c \times p$ as a vector.

For each pixel $y_{i,j}$ in $y$ ($i \in [1, h], j \in [1, w]$), we need to identify similar pixels. The most straightforward approach is to calculate the Euclidean distance between this pixel and others in the image. However, this method is highly susceptible to noise, often leading to incorrect matches. To mitigate this, we first employ the concept of non-local self-similarity by extracting a local patch $p_1 \in \mathbb{R}^{k \times k \times c}$ centered on $y$, and search for $M$ non-local patches $p_m m = 1^M$ that are very similar to $p_1$ within a window of appropriate size $W \times W$, and rank them based on similarity. Euclidean distance is used to measure the similarity between patches. This allows us to assume that pixels at corresponding spatial locations in different non-local patches are similar to some extent. Next, we reshape all non-local patches into column vectors and concatenate them into a matrix of dimensions $k^2 \times c \times M$. At this point, we can consider that the pixels in each row of the matrix are similar. Then, row-by-row, we calculate the sum of the similarities between all pixels in each row and all pixels in the center row. Compared to directly calculating the similarity between pixel $yi, j$ and other pixels, this approach is more robust against noise interference. We extract the $p$ most similar rows to the center row and sort them by similarity. This results in a matrix of size $p \times c \times M$, and we take all the pixels in the first column of the matrix to form a vector of size $1 \times 1 \times c \times p$, which represents the $p$ pixels most similar to $y_{i,j}$ (including $y_{i,j}$). By repeating this process for all pixels in the noisy image, we ultimately obtain a "pixel bank" tensor of size $h \times w \times c \times p$. Once the pixel bank is constructed, we can use a pixel-wise random sampling strategy to sample a large number of instances (a total of $p^{hw}$), which we refer to as "pseudo-instances".

### 3.3 FINE-TUNING THE PRE-TRAINED GAUSSIAN DENOISING MODEL WITH THE NOISE2NOISE FRAMEWORK

In each training iteration, the network randomly samples a pair ($\{C_p^2\}^{hw}$ pairs in total, for each spatial location, we ensure that the pixels sampled twice are different) for training. As shown in Figure 2, during each fine-tuning iteration, the network randomly samples a pair of pseudo-instances (a

total of $C_p^{2^{hw}}$ pairs. Existing methods, such as Noise2Fast (Lequyer et al., 2022) and ZS-N2N (Mansour & Heckel, 2023) are limited to generating four or two predefined sub-images, which can easily lead to overfitting. In contrast, our method aims to generate a large number of training samples, each exhibiting approximately random perturbations in sampling differences, thus enhancing the model's robustness against overfitting. Neighbor2Neighbor (Huang et al., 2021) and Blind2Unblind (Wang et al., 2022b) randomly sample sub-images within a $2 \times 2$ range, but in areas rich in fine details and textures, the pixels in such a small area may not be similar, introducing significant error. Our method leverages the self-similarity of the image to search for similar pixels over a larger area, thereby increasing the similarity of clean content between training pseudo-instances. Table 3 presents the performance of the network fine-tuned with data constructed using Neighbor2Neighbor, ZS-N2N, and our proposed data construction method at different iterations. It can be observed that our method demonstrates the strongest resistance to overfitting, and its performance also surpasses that of Neighbor2Neighbor. This is because the gap between the clean content in the training data constructed by our method is smaller.

Table 3: Using different fine-tuning data construction methods to fine-tune the pre-trained network, we calculate the average PSNR score of the network outputs at different iterations. We mark the first occurrence of the optimal result in **bold**.

| Method | 10 | 20 | 30 | 40 | 50 | 60 | 70 | 80 | 90 | 100 | 200 | 300 |
|---|---|---|---|---|---|---|---|---|---|---|---|---|
| NB2NB | 25.08 | 26.07 | 26.14 | 26.23 | 26.31 | 26.33 | **26.36** | 26.36 | 26.34 | 26.34 | 25.78 | 24.61 |
| ZS-N2N | 26.12 | **26.40** | 26.36 | 26.13 | 25.90 | 25.46 | 24.82 | 24.36 | 23.57 | 22.75 | 18.36 | 16.42 |
| Ours | 26.39 | 26.65 | 26.84 | 26.90 | 26.94 | 27.01 | 27.04 | 27.06 | 27.08 | 27.11 | **27.14** | 27.04 |

## 4 EXPERIMENTS

### 4.1 EXPERIMENTAL DETAILS

**Implementation details.** We use the Neighbor2Neighbor (Huang et al., 2021) model as the pre-trained Gaussian denoiser, which contains only 1.26M parameters. The model is trained on Gaussian noise with $\sigma = 25$. We train the network on 50K images from the ImageNet validation set (Deng et al., 2009). Similar to (Huang et al., 2021), we only select clean images with sizes between $256 \times 256$ and $512 \times 512$ pixels and then randomly crop $256 \times 256$ patches for training. In our implementation, the window size $W$ is 32, the patch size $k$ is 7, and the number of non-local patches $M$ is 16. The number of similar pixels $p$ found at each spatial location is 20. We use the Adam Optimizer to train the network for 10 iterations. The learning rate is set to 0.0001 in synthetic noise experiments and 0.00001 in real-world noise experiments. We implement and train our network using the PyTorch framework on an NVIDIA RTX 3090 GPU.

**Compared methods.** The proposed method is evaluated on several denoising tasks: including Gaussian denoising, Poisson denoising, and real-world RGB image denoising. For Gaussian denoising and Poisson denoising, we compare our method with several state-of-the-art methods including supervised denoising method, DnCNN (Zhang et al., 2017) self-supervised denoising method Neighbor2Neighbor (NB2NB) (Huang et al., 2021), and single image denoising methods, including BM3D (Dabov et al., 2007), DIP (Ulyanov et al., 2018), Self2Self (S2S) (Quan et al., 2020), and ZS-N2N (Mansour & Heckel, 2023). The dataset-based methods (DnCNN, NB2NB) are trained using the same data as our pre-trained Gaussian denoiser. Considering that our method does not require a specific dataset or training the network under a specific noise distribution, we believe our method can also be regarded as dataset-free. For real-world RGB image denoising, in addition to comparing with the aforementioned representative single-image denoising methods, we also compare a single image denoising method MASH (Chihaoui & Favaro, 2024) and three dataset-based self-supervised methods, AP-BSN (Lee et al., 2022), LG-BPN (Wang et al., 2023), and SDAP (Pan et al., 2023), specifically designed for real-world image denoising.

### 4.2 SYNTHETIC NOISE

All methods are tested on the Kodak24 [2] and McMaster18 (Zhang et al., 2011) natural image datasets, as well as the Kvasir (Smedsrud et al., 2021) medical endoscopic dataset. Since the computational cost of running S2S is high, we randomly choose 20 images from the Kvasir dataset to

---

[2] http://r0k.us/graphics/kodak/

Table 4: Average PSNR for Gaussian and Poisson denoising on Kodak24, McMaster18, and KVASIR20. The best and second results are in **bold** and underlined.

| Noise | Method | | Kodak24 | | | McMaster18 | | | KVASIR20 | | |
|---|---|---|---|---|---|---|---|---|---|---|---|
| | | $\sigma$ known? | $\sigma=10$ | $\sigma=25$ | $\sigma=50$ | $\sigma=10$ | $\sigma=25$ | $\sigma=50$ | $\sigma=10$ | $\sigma=25$ | $\sigma=50$ |
| Gaussian | DnCNN | yes | 36.26 | 31.53 | 28.16 | 36.30 | 31.85 | 28.44 | 37.19 | 33.11 | 30.91 |
| | DnCNN | no | 31.88 | 31.53 | 17.95 | 32.47 | 30.36 | 18.73 | 32.61 | 33.11 | 17.74 |
| | NB2NB | yes | 35.81 | 29.70 | 28.45 | 36.06 | 30.36 | 28.97 | 36.98 | 31.77 | 31.32 |
| | NB2NB | no | 32.66 | 29.70 | 20.94 | 33.19 | 30.36 | 21.29 | 33.69 | 31.77 | 21.23 |
| | BM3D | yes | 33.74 | 29.02 | 25.51 | 34.51 | 29.21 | 24.51 | 36.13 | 31.96 | 28.58 |
| | DIP | no | 32.28 | 27.38 | 23.95 | 33.07 | 27.61 | 23.03 | 33.88 | 29.94 | 23.82 |
| | S2S | no | 29.54 | 28.39 | 26.22 | 30.78 | 28.71 | 25.03 | 36.25 | 32.52 | 29.45 |
| | ZS-N2N | no | 33.69 | 29.07 | 24.81 | 34.21 | 28.80 | 24.02 | 35.46 | 31.46 | 28.05 |
| | Ours | no | **34.63** | **30.75** | **27.11** | **34.49** | **31.05** | **27.77** | **37.09** | **33.37** | **29.78** |
| | | $\lambda$ known? | $\lambda=50$ | $\lambda=25$ | $\lambda=10$ | $\lambda=50$ | $\lambda=25$ | $\lambda=10$ | $\lambda=50$ | $\lambda=25$ | $\lambda=10$ |
| Poisson | DnCNN | yes | 31.85 | 30.13 | 27.89 | 32.65 | 30.97 | 28.63 | 33.32 | 32.07 | 30.56 |
| | DnCNN | no | 30.86 | 25.24 | 17.28 | 31.49 | 26.37 | 18.91 | 31.80 | 23.84 | 16.17 |
| | NB2NB | yes | 31.87 | 30.28 | 28.30 | 32.87 | 31.33 | 29.30 | 33.48 | 32.50 | 31.12 |
| | NB2NB | no | 29.62 | 24.82 | 19.94 | 30.42 | 25.81 | 20.69 | 31.01 | 23.26 | 18.66 |
| | BM3D | no | 28.36 | 26.58 | 24.20 | 27.33 | 24.77 | 21.59 | 28.12 | 25.34 | 22.88 |
| | DIP | no | 27.51 | 25.84 | 23.81 | 28.73 | 27.37 | 24.67 | 29.21 | 26.35 | 25.47 |
| | S2S | no | 28.89 | 28.31 | **27.29** | 30.11 | 29.40 | **27.71** | 32.52 | 30.50 | 29.19 |
| | ZS-N2N | no | 29.45 | 27.49 | 24.92 | 30.36 | 28.41 | 25.75 | 31.59 | 30.05 | 27.90 |
| | Ours | no | **30.51** | **28.89** | 26.64 | **31.32** | **29.96** | 27.66 | **33.21** | **31.38** | **29.25** |

Table 5: Denoising PSNR in dB on real-world camera noise.

| Dataset | Dataset-free | | | | | | Dataset-based | | |
|---|---|---|---|---|---|---|---|---|---|
| Method | BM3D | DIP | S2S | ZS-N2N | MASH | Ours | AP-BSN | LG-BPN | SDAP |
| PolyU | 34.66 | 34.75 | 35.97 | 35.17 | 31.97 | **36.71** | 36.49 | 35.76 | 34.15 |
| SIDD | 32.98 | 33.44 | 33.11 | 29.90 | 33.18 | **35.29** | - | - | - |

test. The test images are uniformly center-cropped to generate $256 \times 256$ patches. We consider fixed noise levels of Gaussian noise with $\sigma = 10, 25, 50$ and Poisson noise with $\lambda = 50, 25, 10$.

In Table 4, we present the denoising performance of different methods. It is worth noting that BM3D requires a specific noise level as input. For Gaussian noise, we directly input the actual noise level, while for Poisson noise, we use the estimated noise level based on (Chen et al., 2015). For dataset-based methods, DnCNN and NB2NB, "$\sigma$ known" and "$\lambda$ known" indicate that the training noise distribution matches the testing noise distribution. "$\sigma$ unknown" and "$\lambda$ unknown" indicate that the models trained on Gaussian noise level $\sigma = 25$ are tested on images with noise from other distributions. As shown in the table, dataset-based methods perform better when the training and testing noise distributions are consistent, compared to dataset-free methods. For dataset-free methods, the traditional BM3D method performs well under known noise levels (*Gaussian*) but its performance decreases when the noise level is unknown (*Poisson*). Among deep learning-based methods, DIP lags far behind other methods, while S2S performs well at higher noise levels, although its success is largely due to its ensembling strategy, which often results in overly smooth images. ZS-N2N performs well at lower noise levels but its performance degrades significantly at higher noise levels, which is a result of the noise level differences between the training and testing images caused by its downsampling strategy. Our method not only improves model performance in handling OOD noise but also significantly enhances model performance in handling in-distribution noise (e.g., Gaussian noise, $\sigma = 25$, Kodak24, +1.05 dB; McMaster18, +0.84 dB; KVASIR20, +1.41 dB). Moreover, our method consistently achieves the best or near-best performance in most cases, making it the most robust choice. Please refer to the supplementary materials for visualized results.

## 4.3 REAL-WORLD NOISE

We test all comparison methods on the PolyU dataset (Xu et al., 2018) and the SIDD-Small dataset (Abdelhamed et al., 2018). The SIDD-Medium, PolyU, and SIDD-Small datasets contain 320, 40, and 160 images, respectively. We extract a $256 \times 256 \times 3$ patch from the center of the images in the PolyU and SIDD-Small datasets for testing. The denoising performance of different methods on real camera noise is summarized in Table 5. Since the SIDD-Small dataset is a subset of the SIDD-Medium dataset and the dataset-based methods are trained on SIDD-Medium, we do not present their performance on the SIDD-Small dataset. Unlike synthetic noise, DIP performs exceptionally well when handling real camera noise. Among the data-free methods, our approach achieves the best performance on both datasets, even surpassing the dataset-based methods. Please refer to the supplementary materials for visualized results.

## 5 RELATED WORK

**Supervised denoising methods** mainly rely on paired noise-clean or noise-noise (without clean targets) images for training by convolutional neural networks (CNNs) (Zhang et al., 2017; Chen et al., 2018; Guo et al., 2019; Jia et al., 2019; Lefkimmiatis, 2018) or Transformer networks (Liang et al., 2021; Li et al., 2023; Zhou et al., 2024). They have achieved state-of-the-art performance, but the cost of collecting datasets is high. Although using noise-noise image pairs for training networks relaxes the requirements for training data compared to using noise-clean pairs (Lehtinen et al., 2018), gathering a large number of noise-noise image pairs remains a very challenging task, and sometimes even impossible.

**Self-supervised image denoising methods** use the noisy image itself as its own label, and various constraints or transformations are applied to create a denoised or different noisy version that the model aims to predict. The representative work is blind-spot network based, which includes Noise2Void (Krull et al., 2019), Noise2Self (Batson & Royer, 2019), Laine's (Laine et al., 2019), DBSN (Wu et al., 2020), Blind2Unblind (Wang et al., 2022b), and LG-BPN (Wang et al., 2023). Noisy-as-clean (Xu et al., 2020) and Noisier2Noise (Moran et al., 2020) train the denoising network by adding noise to noisy images. However, they require knowledge of noise distribution. To generate more reasonable training pairs, Recorrupted-to-Recorrupted (Pang et al., 2021) introduces noise with known levels to noisy images, while Neighbor2Neighbor (Huang et al., 2021) obtains training image pairs by downsampling the noisy image. Since these methods rely on the internal knowledge of the training samples, this may lead to decreased effectiveness when faced with noise distributions that differ from the training data distribution.

**Single image denoising methods** rely solely on the to-be-tested noisy image. To date, there have been few methods of this kind, suggesting a significant potential for further exploration. The earliest work is DIP (Ulyanov et al., 2018), which demonstrates the powerful capabilities of neural network architectures, as an image prior for diverse image restoration tasks. Self2Self (Quan et al., 2020) trains with dropout on Bernoulli sampling instances of the input image. In order to efficiently acquire pairs of noisy images, Noise2Fast (Lequyer et al., 2022) introduces the checkerboard downsampling to generate training image pairs. Inspired by Noise2Fast, ZS-N2N (Mansour & Heckel, 2023) proposes a new image downsampling technique to obtain training image pairs. These single-image denoising methods do not require prior training under specific noise conditions, offering greater flexibility and versatility in handling various types of unseen noise. MASH (Chihaoui & Favaro, 2024) introduces a shuffling technique to weaken the local correlation of noise, which in turn yields an additional denoising performance improvement.

**Test-time adaptation.** In recent years, test-time adaptation (TTA) methods (Liang et al., 2024; Chen et al., 2022; Wang et al., 2021) have been introduced to mitigate domain shift by updating pretrained models online using test data. EATA (Niu et al., 2022) proposes an active sample selection criterion to identify reliable and non-redundant samples in order to minimize entropy loss during test-time adaptation. MEMO (Zhang et al., 2022) proposes a probabilistic and adaptive testing setup that can be used for any model. LAME (Boudiaf et al., 2022) proposes using a Laplace-adjusted maximum likelihood estimation objective to address the problem that previous methods sometimes fail catastrophically when their hyperparameters are not chosen for the test scenario. SRTTA (Deng et al., 2023) focuses on the image super-resolution task and addresses the degradation shift issue. SS-TTA (Fahim & Boutellier, 2023) synthesizes additional Gaussian noise training images at test time to refine the result from the pre-trained denoised model. The adaptation capability of the pretrained model improves in the case of Gaussian noise but fails to generalize to other types of noise, such as Poisson noise.

## 6 CONCLUSION

In this paper, we propose a test-time adaptation framework for image denoising to quickly mitigate the domain shift problem in image denoising. We first train a Gaussian denoiser with deep denoising prior by using a self-supervised approach. Then, we propose a self-similarity-based method to construct a pixel pool for each test image, from which pseudo-instances are sampled for fine-tuning, reducing the gap between the clean contents of training sample pairs. Extensive experiments on both in-distribution synthetic noise, out-of-distribution noise, and real-world noise demonstrate that our method can quickly adapt the denoising model for each image and recover high-quality images.

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
