# OpenReview forum: "Deep Denoising Prior: You Only Need a Deep Gaussian Denoiser"
_ICLR.cc/2025/Conference — ICLR 2025 Conference Withdrawn Submission_

### Official Review · Reviewer_UK9P · 2024-10-18

**Soundness:** 3
**Presentation:** 3
**Contribution:** 3
**Rating:** 5
**Confidence:** 4

**Summary:**

This paper proposes a test-time adaptation method for image denoising to address the noise discrepancies between training and testing images. The method consists of three steps: first, a Gaussian denoising network is trained; second, for each testing image, self-similarity is utilized to construct a pixel bank and generate pseudo noise-noise pairs; finally, the Gaussian denoising network is fine-tuned on the generated noise-noise pairs. The authors validate the performance of the proposed method on both synthetic noise and real-world noise. Additionally, they conduct a comprehensive ablation study for each parameter.

**Strengths:**

1. The idea of using self-similarity to construct pseudo noise-noise pairs is novel and appears valid.
2. The authors have conducted a comprehensive ablation analysis.
3. The method demonstrates superior performance compared to other single-image-based denoising methods.
4. The proposed test-time adaptation method is less time-consuming than other single-image-based approaches.

**Weaknesses:**

I believe that the comparison with single-image-based denoising methods is somewhat unfair, as the proposed model requires pretraining on a dataset with Gaussian noise, whereas the single-image-based methods do not require that pretraining. Additionally, the results for SDAP [1] are inconsistent with those reported in the original paper. The claim that this method surpasses dataset-based methods is questionable. It would also be beneficial to include the results for DND in the analysis. The paper should focus more on real-world noise, as the proposed method aims to address the discrepancies between synthetic Gaussian noise and real-world noise; therefore, experiments involving synthetic noise are of lesser importance.

[1] Pan, Yizhong, et al. "Random sub-samples generation for self-supervised real image denoising." Proceedings of the IEEE/CVF International Conference on Computer Vision. 2023.

**Questions:**

1. I believe that using self-similarity to construct pseudo noise-noise pairs is an intriguing idea. Why not develop a dataset-based denoising method using these training pairs, similar to NB2NB?
2. For the SDAP method, the authors should provide comparisons with SDAP(S)(E).
3. Also, could you conduct a comparison with the LAN [2] method, which is also a noise adaptation approach?

[2] Kim, Changjin, Tae Hyun Kim, and Sungyong Baik. "LAN: Learning to Adapt Noise for Image Denoising." Proceedings of the IEEE/CVF Conference on Computer Vision and Pattern Recognition. 2024.

---

### Official Review · Reviewer_pK9U · 2024-11-01

**Soundness:** 3
**Presentation:** 4
**Contribution:** 3
**Rating:** 5
**Confidence:** 3

**Summary:**

The paper introduces a test-time adaptive denoising framework where a pre-trained Gaussian denoiser is fine-tuned during inference to handle diverse noise types. This adaptation is achieved by leveraging self-similarity-based pseudo-instance construction. Authors leverage diverse training techniques to train a denoising model such as Noise2Noise with their idea. The experiments demonstrate significant improvements in both denoising performance and runtime efficiency, compared to other self-supervised denoisers.

**Strengths:**

- Strong performance in removing Gaussian (in-distribution) and Poisson (out-of-distribution) noise, outperforming other self-supervised methods.
- The method enables test-time adaptation while maintaining computational efficiency.
- Self-Similarity Based Pseudo-Instance Construction (SSPC) is clear and intuitive.
- Experiments are well-conducted and ablation studies are done.
- The paper is well-written and easy to follow.
- Authors provided the code to reproduce the results.

**Weaknesses:**

- The rationale for pre-training on Gaussian noise is unclear. The authors claim that Gaussian noise provides a general deep denoising prior (as highlighted in the title), but no mathematical justification is offered beyond experimental results. Why does it have to be Gaussian? Are there any connections with proposed fine-tuning method?
- The generalization capability is uncertain, especially given the need to empirically determine optimal hyperparameters (e.g., learning rate, number of similar pixels). For instance, while 20 similar pixels were used for SSPC, Table 5 in the Appendix shows that lower numbers of similar pixels perform better in real-world cases.
- The method's applicability to real-world scenarios remains questionable, as the performance (e.g. SIDD benchmark) still falls short compared to simply supervised methods. How could this be further developed for practical use?

If these weaknesses are addressed, I will improve my score.

**Questions:**

Please read the Weaknesses section for main questions. These are minor.

- What criteria were used to select different learning rates for synthetic and real-noise experiments? Can you provide ablation studies on this choice?
- If the framework is network-agnostic, the method might perform better if adapted with recent architectures. Are you planning to apply your method to other architectures to validate its general applicability?
- What are the weaknesses of the proposed method? Are there any failure cases for denoising?
- The term 'synthesis noise images' in Figure 2 appears to be a typo of 'synthetic noise images'?

---

### Official Review · Reviewer_ZMny · 2024-11-02

**Soundness:** 2
**Presentation:** 2
**Contribution:** 2
**Rating:** 3
**Confidence:** 5

**Summary:**

This paper presents a self-supervised technique for training image denoisers. The authors propose training denoisers in a noise-to-noise manner using pairs of noisy pseudo-instances. In this approach, one pseudo-inverse serves as the network input, while the other is used as a target image. The pseudo-instances are generated from the original noisy image through non-local patch matching. The training process consists of two stages: an initial pre-training for Gaussian image denoising with sigma 25, followed by an adaptation phase. In the adaptation phase, given a noisy image with an unknown noise distribution, the denoiser is adjusted to handle that specific noise by constructing noisy pseudo-instances and training the denoiser on pairs of these instances.

**Strengths:**

Adapting denoisers to out-of-distribution noise is an important challenge, as denoisers trained on Gaussian noise may not perform well on other types of noise, such as real-world or microscopic noise. The idea of leveraging the self-similarity within images for self-supervised training is, in my opinion, quite interesting. Before the deep learning era, patch matching was widely used in image-denoising algorithms (e.g., [1]-[3] and others). This suggests that exploring self-similarity in the context of self-supervised training could be a valuable approach.

[1] Dabov, Kostadin, et al. "Image denoising by sparse 3-D transform-domain collaborative filtering." IEEE Transactions on image processing 16.8 (2007): 2080-2095.

[2] Gu, Shuhang, et al. "Weighted nuclear norm minimization with application to image denoising." Proceedings of the IEEE conference on computer vision and pattern recognition. 2014.

[3] Mairal, Julien, et al. "Non-local sparse models for image restoration." 2009 IEEE 12th international conference on computer vision. IEEE, 2009.

**Weaknesses:**

While this manuscript has its strengths, it also contains a significant weakness. The use of self-similarity for self-supervised training of denoisers is not a novel concept. The approach of constructing noisy pseudo-instances using patch-matching was introduced in [4]. Although the algorithm in this manuscript differs from that in [4], both algorithms rely on the same core idea: leveraging self-similarity through patch-matching.

In [4], the pseudo-instances are created by replacing the image patches with non-local patches. In contrast, the authors of this manuscript propose a two-stage algorithm for forming pseudo-instances. In the first stage, they apply patch-matching. In the second stage, they use the matched non-local patches to build feature vectors for each image pixel. Then, these vectors are used to identify similar pixels, and pseudo-instances are created by replacing the original pixels with similar ones.

Since this manuscript and [4] are built on the same core concept, it would be valuable for the authors to clarify the distinctions between the two methods and explain how and why these differences result in improved performance compared to [4]. For example, a notable difference is that [4] requires a burst of images to create pseudo-instances, whereas the current manuscript can generate pseudo-instances from a single noisy image.

[4] Gregory Vaksman and Michael Elad. Patch-craft self-supervised training for correlated image denoising. In Proceedings of the IEEE/CVF Conference on Computer Vision and Pattern Recognition, pp. 5795–5804, 2023.

**Questions:**

- I am curious why the authors chose to pre-train the denoiser in a self-supervised manner. Supervised training, which uses pairs of noisy-clean instances, typically performs better and is often simpler to implement.

- Test-time adaptation is particularly valuable for handling unknown noise distributions. Standard i.i.d. Gaussian and Poisson noise can often be managed efficiently by supervised denoisers. Therefore, it would be beneficial to conduct additional experiments with noise types that are more challenging to estimate or simulate, such as real-world camera noise.

---

### Official Review · Reviewer_YFAN · 2024-11-07

**Soundness:** 2
**Presentation:** 2
**Contribution:** 1
**Rating:** 3
**Confidence:** 5

**Summary:**

This work proposes methods to enhance the self-supervised training capability for deep denoisers by proposing a pipeline with pretraining with Gaussian noise and by proposing a method to generate pseudo-instanced with the well-known Noise2Noise loss. The methods were evaluated on a number of popular benchmarks such as Kodak24, McMaster18, KVASIR20 (synthetic noises) as well as SIDD, PolyU (real noises), demontrating superior performance over some of the prior arts such as BM3D, DIP, S2S, ZS-N2N, MASH and NB2NB.

**Strengths:**

- Proposed pseudo instance generation method looks interesting.
- The proposed method yielded SOTA results over some of the prior arts.

**Weaknesses:**

- Non local search for pseudo instance does not always give the intended images that satisfies the assumption of Noise2Noise (e.g., image residual bias). However, there is no theoretical consideration for it in this work.
- Investigation on pretrained models in this work looks quite limited with a couple of examples while the claims look quite general.
- Experiments in this work do not look comprehensive to demonstrate that the proposed method is indeed a SOTA. More recent methods, benchmarks, and situations should be investigated.
- Some of the experiments do not look fair. For example, Table 5 compared the proposed method (with the pretrained model with massive data) with other methods that only used the test image without using any pretrained model. Similarly, in Table 4, the proposed method was compared with the same methods for both Gaussian and Poisson cases, but the method should be compared with more appropriate methods for Poisson case.

**Questions:**

- Please address the concerns in the weakness section.
- LPIPS is a good metric to compensate for PSNR and it will be interesting to use it as another metric.

---

### Note · Authors · 2024-11-14

I have read and agree with the venue's withdrawal policy on behalf of myself and my co-authors.